# Questioning Gender and Sexual Identity in the Context of Self-Concept Clarity, Sense of Coherence and Value System

**DOI:** 10.3390/ijerph191710643

**Published:** 2022-08-26

**Authors:** Julia Jastrzębska, Magdalena Błażek

**Affiliations:** 1Faculty of Health Sciences, Medical University of Gdansk, 80-210 Gdansk, Poland; 2Department of Psychology, Medical University of Gdansk, 80-210 Gdansk, Poland

**Keywords:** gender identity, sexual identity, self-concept clarity, sense of coherence, value system

## Abstract

Sexual and gender identity is a fundamental part of one’s overall identity and plays an important role in human functioning. Questioning one’s sexuality associated with low level of self-concept clarity, certainty, consistency and stability with regard to the individual’s beliefs about oneself, can affect their sense of coherence and value system. The aim of the study was to compare heterosexual and cisgender people with non-heteronormative and non-cisgender people regarding their attitudes and the way they perceive significant personal values. It was assumed that non-heterosexual and non-cisgender individuals would have lower self-concept clarity and lower sense of coherence, and that among them such values as openness to change and transcending Self would be dominant. The study was conducted on a group of 337 individuals aged 18 to 30. The participants filled out four self-report online questionnaires. Self-concept clarity was found to be connected with a greater tendency to question one’s sexual and gender identity. The results also indicate differences between heterosexual/cisgender participants and non-heterosexual/non-cisgender participants in terms of the degree of self-concept clarity and sense of coherence. Non-heteronormative and cisgender individuals show a greater tendency to question their identity and have lower self-concept clarity, which may lower their sense of coherence.

## 1. Introduction

Identity is a concept that denotes the organisation of information that makes up self-perception, including a system of self-knowledge, a sense of one’s own existence, or a set of descriptions that one considers characteristic of oneself [1,2]. Due to the complexity of the identity structure, the literature on the subject describes numerous terms referring to different identity-related phenomena, focusing on different aspects of identity. With regard to experiencing own identity by an individual, terms such as “I”, denoting self-awareness, or “sense of identity”, emphasising the subjective nature of identity, are used [3]. Regardless of the definition, the concept of identity refers to the integral phenomena of human functioning as an individual defining oneself on many levels [2], who acts both as an object and a subject in this process [4]. The complexity of identity is reflected in the way of mapping reality, interpreting oneself and one’s position in the context of relationships with people, things or values [5]. The content of identity is shaped by experiences, values and meaning assigned by the individual to them [6]. Identity, thus, plays a very important role in life. It is one of the most important factors shaping behaviour, both in the context of mental and social functioning. It regulates cognitive, emotional, motivational and behavioural processes and determines the individual’s place in time and space [2]. Striving for individuality and trying to stand out against one’s social background—characteristics typical of human beings—is a fundamental manifestation of the Self, which helps to adapt to the environment and sustain mental health. Consequently, as a result of a violation of one’s sense of identity, a person may experience a disruption of their psychological well-being.

Although identity may appear to be relatively constant, it is important to emphasise that its different areas are shaped throughout human life [7,8], as a natural consequence of personal development. One of the most important periods in life, during which identity is formed most actively on different levels, is the so-called “emerging adulthood” [9], understood as a transition stage between adolescence and early adulthood, and characterised by instability and exploration. It is a time of frequent life changes, trying new challenges, experimenting and experiencing new things, which is the basis of self-definition in the field of relationships and world-view [7,10]. It can be said that the formation of identity through its questioning is an essential part of this stage of development and, as a process, depends on a number of factors and the context in which it takes place, including current emotional, cognitive and social resources. What appears to be of particular importance is the environment in which an individual is functioning during the process of identity formation, due to high sensitivity to the feedback received from that environment about oneself [11].

One of the most significant areas in which identity is formed is sexuality, encompassing gender and sexual identity as elements that determine sexual functioning, but also reach far beyond that sphere. Gender identity refers to the inner sense of one’s gender and the way it is named. Sexual identity, on the other hand, is a concept related to experiencing oneself as a person feeling sexual drive, which leaves space for the subjectivity, dynamism and fluidity of human experience [12], much broader than psychosexual orientation, which denotes a permanent emotional and sexual attraction towards a specific sex. As gender identity has for long been the most widely used basis for the classification of people, its relevance to everyday life is fundamental. Due to the fact that the impact of identity on one’s life in the context of sexuality is significant, the consequences of the increasing difficulties people have in defining it can significantly affect their overall functioning. The reasons for mounting doubts and the search for one’s identity can be found not only in the processes that are natural consequences of human development [7], but also in social and cultural factors. Social pressure and expectations with regard to self-determination may prompt one to question assumptions about oneself so that they are in line with these expectations, which may negatively affect the psychological well-being of people with struggle with self-determination.

What is closely related to questioning one’s own identity, understood as lack of certainty when it comes to defining oneself or searching for one’s identity, is self-concept clarity [13]. Self-concept clarity, as part of general identity, is the highest level of personality organisation, which defines the degree to which one’s perception of oneself is clearly and confidently formulated, internally consistent and stable over time [13,14]. A solid and well-grounded sense of identity may appear to be advantageous from the point of view of adaptation. On the other hand, individuals with lower self-concept clarity and greater flexibility in this regard, may sometimes function better in the society, as they adapt to their environment with greater ease [15]. However, it should be stressed that prolonged inability to organise one’s own experiences and to define oneself in important areas of life, causing deep insecurity with regard to the Self, may contribute to increased anxiety and disturb the sense of security of the individual with such difficulties, disturbing their well-being and social functioning [2].

The inability to define one’s own identity significantly affects other aspects of human functioning, related to the perception of reality and assessing events in terms of value systems. Therefore, it is worth to analyse the correlations between the phenomenon of questioning one’s identity and low self-concept clarity on the one hand, and the sense of coherence and value system, on the other.

The sense of coherence is one of the so-called salutogenic factors identified by Aaron Antonovsky (1979) and denotes the global orientation of an individual, reflecting the degree to which they have a stable, yet at the same time dynamic, feeling of certainty that (1) the stimuli from the internal and external environment are structured, predictable and comprehensible (comprehensibility), (2) they have access to the means allowing to meet the demands of these stimuli (manageability), (3) these demands are a challenge worth the effort and engagement (meaningfulness). The sense of coherence, thus, determines how reality is perceived, felt and understood, and is related to the resources of the individual. People with a high sense of coherence are highly motivated and active; they can cope with difficult situations and are able to adapt to their environment. On the other hand, people with a low sense of coherence may experience difficulties in coping with stressors [16,17]. Assuming that the sense of coherence is an orientation shaped under the influence of numerous factors (e.g., experiences shaping one’s beliefs) and having an impact on various aspects of life, it can be said that it remains in close relation to human identity.

Another aspect––important from the point of view of human functioning in the context of one’s beliefs and activity––is the value system. Values can be defined as motivational constructs that reflect what is important to individuals. In other words, values refer to desirable goals that motivate people to take certain actions and serve as standards, shaping identity and constituting one of the fundamental aspects of the concept of oneself [18,19]. Shalom Schwartz (2006)––the author of the most widely used value system concept––proposed a comprehensive catalogue of 10 types of values of varying degrees of importance to different people. These include: (1) security––a sense of safety and order; (2) power––social status and a degree of control over other people and resources; (3) achievement––personal success achieved in accordance with social standards; (4) hedonism––focus on pleasure and satisfaction of one’s needs; (5) stimulation––a search for originality and variety in life; (6) self-direction––independence of one’s thoughts and actions, autonomous decision making; (7) universalism––concern for the well-being of people, care for the environment, striving for justice, wisdom and peace; (8) benevolence,––care for one’s nearest and dearest resulting from the feeling of love and friendship; (9) tradition––respect and acceptance of cultural or religious rituals, and (10) conformity––self-discipline, limiting one’s own aspirations for the benefit of others and respect for the older generations. The values listed above form a circular model, in which they complement or contradict one another. Schwartz also proposed an analysis of the value system based on two dimensions: openness to change versus conservation and self-enhancement versus self-transcendence. The first dimension is related to the conflict between such values as self-direction and stimulation versus security, adaptation and tradition. The second dimension describes the conflict between achievement and power versus universalism and benevolence. Hedonism, in turn, is related to both openness to change and self-enhancement [18,19,20,21,22,23]. In addition, building on the motivational aspect of this theory, Schwartz (2006) made a distinction that assumes that some values stem from fear (tradition, adaptation, security, power), while others do not (universalism, benevolence, stimulation, self-direction, hedonism) [24].

Based on Schwartz’s theory, it could be assumed that for people who question their sexuality in the context of the difficulties they experience in this regard, the values important to them will be those that stem from fear of social stigma and exclusion; Such individuals will find themselves in the dimensions on the continuum of openness to change and self-transcendence.

He discussed issues are interrelated, and the way an individual perceives and defines oneself affects their experiences and their interpretation, as well as the activities they undertake and the way they function in a broader perspective. The main objective of this paper was to compare:(1)heterosexual and cissexual individuals with non-heterosexual and non-cissexual individuals, and(2)individuals who defined themselves in the context of their sexuality with individuals who are in the process of discovering their identity or are questioning it, in terms of the sense of coherence and declared value system treated as a result of experiences related to sexuality.

It was assumed that:(1)low self-concept clarity correlates with a tendency to question one’s sexuality;(2)individuals questioning their identity with low self-concept clarity will have a lower sense of coherence than individuals with high self-concept clarity who identify themselves with a particular psychosexual orientation and gender;(3)values constituting the metatypes of openness to change and self-transcendence will be dominant among individuals questioning their identity.

## 2. Materials and Methods

### 2.1. Participants

The total number of *N* = 337 persons were examined; the largest group were persons with secondary education (113) and higher education (144). The respondents were aged 18 to 30, the mean age being M = 21.41 (SD = 3.14).

Due to the large variety of answers and need for simplification, the category “Other” covers every person who, when specifying their sexual identity, went beyond the specified answer options (“Heterosexual”, “Homosexual”, “Bisexual”, “Asexual”, “I have not yet specifed my sexual orientation”) and used the space for the open answer. People who defined themselves as pansexual (24) and demisexual (8) created quite a large group. Some of the respondents answered the question about their sexual identity in a descriptive manner. In this group of respondents there were answers such as: “I do not get attached to any psychosexual orientation, I do not want it to limit me.”, “I have the impression that my orientation changes depending on my mood and I cannot explain it.”, “I am pretty sure that I am heterosexual, but I don’t exclude that I may be bisexual.”, “At the moment I am close to asexuality but I am not 100% sure.”. In addition, several people used 2 labels to define their sexual identity, for instance” “bisexual and heteromantic”, “pansexual and demisexual”, “heterosexual and demisexual”, “demisexual and sapiosexual” (See Table 1.).

### 2.2. Methods

Three self-report questionnaires and additional questions were used for the purpose of this study:

Self-Concept Clarity Scale (SCC) (Błażek, Besta, Kaźmierczak, 2019) [25] containing 12 statements directly describing an individual’s beliefs about the structure of their Self and clear sense of identity, including statements such as: “My perceptions of myself seem to change very often”, “My perceptions of myself are often contradictory”, to which the person should respond indicating the extent to which they agree with a given statement on a scale from 0 to 6.

The SOC-29 Life Orientation Questionnaire (Antonovsky, 1993) [26], which determines the degree of the sense of coherence and its individual components, consisting of 29 self-report questions to which one has to respond using a 7-point scale. Examples of the questions include: “When you talk to people, do you have the feeling that they do not understand you?”, “How often do you have the feeling that what you do every day is not very meaningful?”.

The Polish version of the Portrait Values Questionnaire PVQ (2011) [27] developed by S. Schwartz, adapted by Cieciuch and Zaleski, is used to measure values and consists of 40 short descriptions of people. The respondent’s task is to assess how they relate to the person described, using a 6-point scale (from “not at all like me” to “very much like me”). The questionnaire allows to estimate the individual’s preference for each of the 10 types of values.

Additional questions, used to determine the degree to which one questions their own sexuality (the so-called questioning degree index), address the individual experience of own sexuality, encompassing such issues as: coming out, acceptance of one’s identity or support from family members and friends (Appendix A).

### 2.3. Procedure

The study had been conducted through social media. The participants had been recruited using the Facebook platform. A link to the questionnaire was posted in groups dealing with such topics as psychology, sexology and the LGBTQ+ community. Above mentioned groups are a space for their users to exchange views and thoughts. They also aim to support and integrate people from the LGBTQ+ communities, which allowed to recruit the required number of non-heteronormative respondents. Each of the groups also included cisgender and heterosexual individuals. In order to join such a group every member had to agree to the rules applicable in it. These groups are monitored by their administrators and moderators which have agreed to post the link to the study on them.

The participants were asked to fill out online questionnaires. At the beginning of the study the participants were given information about purpose, subject and course of the study and that participation in it was completely anonymous and voluntary. After getting acquainted with this information they agreed to participate in the study, which allowed access to further parts of the questionnaire. The nature of the media used and the voluntary participation in the study, made it possible to avoid careless respondents.

## 3. Results

Data collected from 337 individuals was analysed. Statistical analyses were performed using the IBM SPSS Statistics 27.0.1.0 (Poland, Gdansk).

### 3.1. Self-Concept Clarity and Questioning One’s Own Identity

Pearson’s r correlation analysis showed a slight positive correlation between the self-concept clarity score and the questioning degree index (r = 0.379, *p* < 0.01) (the lower the self-concept clarity, the higher the tendency to question one’s own sexuality) (Table 2).

### 3.2. Self-Concept Clarity, Questioning Own Identity and Sense of Coherence

The model was found to be statistically significant F(2334) = 143.809, *p* < 0.001. SCC score is a significant predictor (β = 0.653, *p* < 0.05), which means that the higher the self-concept clarity, the higher the sense of coherence (Table 3).

### 3.3. Comparison by Gender Identity

The Table 4 presents only those results that were found to be statistically significant. Statistically significant differences were found for cisgender vs. non-cisgender respondents in the areas of self-concept clarity, sense of coherence, questioning degree index, values such as tradition and universalism, and the dimension openness to change versus conservation. Cisgender respondents had higher self-concept clarity and sense of coherence and tended to question their sexuality less frequently. There were no statistically differences in the result of: self-transcendence *t*(322) = −1.039, *p* > 0.01 between cisgender (M = 47.6, SD = 7.2) and non-cisgender (M = 49.0, SD = 6.0); openness to change *t*(322) = −0.047, *p >* 0.01 between cisgender (M = 42.0, SD = 8.1) and non-cisgender (M = 42.1, SD = 7.1).

### 3.4. Comparison by Sexual Identity

The Table 5 presents only those results that were found to be statistically significant. Statistically significant differences were found for heteronormative vs. non-heteronormative people in the areas of self-concept clarity, sense of coherence, questioning degree index, and values such as conformity, tradition, self-direction, stimulation, security, and on the dimension of openness to change versus conservation. Non-heteronormative individuals were more likely to question their sexuality. There were no statistically significant differences in the result of: self-transcendence *t*(310) = 0.300, *p* > 0.01 between heteronormative (M = 46.7, SD = 7.7) and non-heteronormative (M = 47.3, SD = 7.2); openness to change *t*(310) = 1.942 *p* > 0.01 between heteronormative (M = 40.8, SD = 8.7) and non-heteronormative (M = 43.0, SD = 7.9).

## 4. Discussion

Based on the results, self-concept clarity was found to be connected with a greater tendency to question one’s sexual and gender identity. It has to be noted, however, that this correlation may work both ways. Individuals with low self-concept clarity, whose perceptions about oneself are generally inconsistent, unstable and insecure, may be prone to questioning every area of their identity, including sexuality. On the other hand, people who question their sexuality as one of the most important and fundamental areas of their lives, who are not fully able to define themselves and who experience confusion, may also have difficulties with self-determination in terms of their identity beyond this area, and may become prone to question all information about themselves.

Regarding the hypothesis that assumes the existence of differences between individuals with a specifically defined gender identity and those who have questioned or are in the process of questioning and defining their gender identity, the study has found that cisgender people (identifying as female or male, with no discrepancy between their assigned gender at birth and gender identity) have higher self-concept clarity, higher sense of coherence and are less likely to question their gender identity than non-cisgender people (whose assigned gender at birth is different from their gender identity, as well as those who go beyond the binary gender classification when defining themselves). This may be due to the fact that very often the process of self-definition for transgender people is a very difficult and complicated experience that entails psychological costs, also as a result of the lack of understanding and acceptance by the society. Such individuals often have difficulties in accepting themselves, experience severe gender dysphoria, which is chronic distress and suffering related to the discrepancy between their gender identity and their assigned gender at birth, and struggle with internal conflict and lack of support [28,29]. They are also often forced to express themselves contrary to how they feel in order to fit into their environment. Apart from questioning one’s identity, all these experiences can significantly affect their perception of themselves, people and the world at large, lowering their sense of coherence and self-esteem, and significantly hindering their functioning in all areas of life.

Comparison of cisgender and non-cisgender people in terms of their value systems showed that universalism, understood as concern for the well-being of loved ones, the feeling of love and friendship, is more important for non-cisgender people than for cisgender people, which may be related to the fact that, faced with rejection due to their identity, non-cisgender people experience greater fear of losing their loved ones and care for them more or––when they have already experienced rejection––appreciate intimate relationships to a greater extent. Compared to non-cisgender people, tradition, understood as respect for the rituals and ideas of one’s own culture or religion, was found to be more important for cisgender people. This may be due to the fact that the cultural background of the non-cisgender respondents or the religion they practise do not fully recognise and accept individuals who identify themselves other than in terms female or male, and people who go beyond the stereotypical canons are often perceived in negative terms. It should be stressed, however, that the differences between the compared groups in terms of values are not large, which can be explained by the fact that an individual’s value system is shaped by numerous factors and that there are some universal values independent of sexuality or its experience.

Similar results were obtained when comparing heteronormative and non-heteronormative persons. For heterosexual individuals, values such as security (i.e., a sense of safety, harmony and social order in personal terms, within one’s family and country) and tradition were found to be more significant than for non-heteronormative persons. One might expect the opposite, namely that security should be a more important value for non-heteronormative persons, as they are in a sense more exposed to exclusion, misunderstanding, and even violence by people who do not accept their identity, which is less frequently experienced by heterosexual persons. However, this may be due to the fact that non-heterosexual individuals are somehow accustomed to a lack of harmony or a sense of security, hence state security and social order are not so important to them because of the negative experiences they have experienced on the part of the society. It is also worth noting that heterosexual people scored higher in terms of conservation, which means that for such individuals it is more important to maintain the existing order and that they are more reluctant to changes and self-limitations compared to non-heteronormative people. This is understandable given that non-heteronormative individuals are more prone to question and search for their identity, and hence demonstrate a high degree of openness. On the other hand, values such as self-direction and stimulation were found to be less important for heterosexual people, which may indicate that non-heterosexual people who question their heterosexual identity are more independent in terms of their thoughts and actions and are more likely to seek novelty, which is also associated with greater openness to experience.

The results comparing heteronormative and non-heteronormative individuals in terms of self-concept clarity and sense of coherence are similar to those comparing individuals questioning and not questioning their gender identity. Heterosexual respondents scored higher in terms of questioning degree index (they are less likely to question their identity), self-concept clarity and sense of coherence. The results of non-heterosexual respondents may be explained by the very fact of questioning their heterosexual orientation, which is why their sense of identity may be less stable compared to heterosexual individuals. One should also take into account the experiences of non-heterosexual individuals in the context of social functioning and the potential lack of acceptance both from strangers and persons close to them.

What is important in terms of the obtained results is that the study involved young people, who may still be in the process of developing and forming their identity in general [10]. In addition, there were disproportions in terms of the number of respondents assigned to given categories of gender or sexual identity (predominance of women), which can be viewed as a reflection of reality, and which might have had an impact on the results. No significant differences have been found between the respondents questioning and those not questioning their sexual identity, which may also be related to the significant disproportion between these groups.

When analysing phenomena related to human sexuality, it should be remembered that sexology is currently one of the most rapidly developing scientific fields, and the approach to human sexuality is undergoing dynamic changes. The development of language as a tool for defining oneself in the context of one’s sexual and gender identity in the context of expanding horizons of human experience, may, in a sense, provoke or even force one to question one’s identity. Widespread access to knowledge allows people to study different issues, and the contradictions in the views on human nature presented by various authorities and the great number of possible explanations do not facilitate the search for oneself. Social pressure and expectations towards self-determination often seem unnecessary and harmful to individuals. Nevertheless, paraphrasing the words of Prof. Piotr Oleś, it seems that it is better to have a specific sexual and gender identity than not to have one. It is very important to analyse the phenomena in question in the context of social transformations. Both the sense of coherence and the value system may be affected not so much by the fact of questioning one’s sexuality, but rather by all the experiences associated with it, such as the lack of acceptance and support from the society, which sometimes forces one to adapt to the environment and reject one’s personal feelings. LGBTQ+ individuals experience a high sense of anxiety, which can modify their perception of themselves and the world around them. Their sense of coherence may be affected by a number of different factors, including resources for coping with difficulties, which have not been examined in this paper. Low self-concept clarity may––on the one hand––have a negative impact on psychological well-being, but––on the other––it may facilitate adaptation. It is difficult to conclude with certainty whether identity variability and inconsistency are attributes of adaptability or maladaptability [2]. The question is what is more desirable by contemporary people––the desire for stability or flexibility. Nevertheless, given the significant impact of the aspects discussed on the broadly understood human health and functioning as well as due to insufficient data in this regard, further research is recommended.

## 5. Conclusions

1.Non-heteronormative and cisgender individuals show a greater tendency to question their identity and have lower self-concept clarity, which may lower their sense of coherence.2.Socio-cultural context may have a significant impact on the course of the process of questioning one’s sexual and gender identity.

## Figures and Tables

**Table 1 ijerph-19-10643-t001:** Characteristics of the respondents in terms of gender and sexual identity.

	*n*	%
Gender identity		
Women	224	66.5
Men	53	15.7
Transgender	45	13.3
Individuals who did not identify themselves	15	4.5
Sexual identity		
Bisexual	92	27.3
Homosexual	52	15.4
Heterosexual	112	33.2
Asexual	12	3.6
Other	42	12.5
Individuals who did not identify themselves	27	8

**Table 2 ijerph-19-10643-t002:** Questioning degree index––SCC score correlation.

	Questioning Degree Index
SCC score	0.379 **

** *p* < 0.01.

**Table 3 ijerph-19-10643-t003:** Regression forthe SOC-29 total score.

	*B*	β	*t*	*p*
Constant	56.314		12.196	0.000
SCC score	1.046	0.653	15.055	0.000
Questioning degree index	0.546	0.066	1.512	0.132
Adjusted R^2^ = 0.459				

**Table 4 ijerph-19-10643-t004:** Differences between cisgender and non-cisgender people. (Cisgender respondents were defined as individuals who identified themselves as female or male and whose gender identity was consistent with their birth sex. Non-cisgender respondents were defined as individuals who identified as transgender and/or non-binary.).

	Cisgender		Non-Cisgender		*t*(322)	*p*
	*M*	*SD*	*M*	*SD*		
SCC score	37.5	16.5	29.8	13.8	−3.67	0.000
SOC-29 (total score)	103.8	26.2	90.4	21.8	3.65	0.000
tradition	10.9	3.9	9.4	3.3	2.82	0.050
universalism	28.9	4.7	30.3	4.3	−2.05	0.041
conservation	43.3	10.1	39.9	8.8	2.33	0.021
Questioning degree index	13.9	2.9	11.0	2.5	7.8	0.000

**Table 5 ijerph-19-10643-t005:** Differences between heteronormative and non-heteronormative people.

	Heteronormative		Non-Heteronormative		*t*(310)	*p*
	*M*	*SD*	*M*	*SD*		
SCC score	40.7	16.4	31.8	14.3	3.90	0.000
SOC-29	108.8	27.1	92.6	23.0	4.36	0.000
conformity	14.3	4.0	11.9	4.1	−4.20	0.000
tradition	12.2	4.3	9.4	3.0	−5.39	0.000
self-direction	18.5	3.5	19.6	3.4	2.16	0.032
stimulation	10.5	3.7	11.5	3.6	2.07	0.040
security	19.5	4.3	18.2	4.5	−2.17	0.031
conservation	46.0	10.3	39.5	8.9	−4.79	0.000
Questioning degree index	15.6	2.1	12.5	2.9	−8.74	0.000

## Data Availability

The data presented in this study are available on reasonable request from the corresponding author.

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
