# Peer review of "Questioning Gender and Sexual Identity in the Context of Self-Concept Clarity, Sense of Coherence and Value System"

_ijerph, 2022, doi:10.3390/ijerph191710643_

Round 1

Reviewer 1 Report

This paper explores an interesting sample and topic. Sexual identity, especially in sexual minority populations, is a growing area of research interest, and this project offers some useful information about individual differences and values in these populations.

Much more information is needed in the Method. In particular, additional information about informed consent would be welcome. Information about recruitment and sampling is critical to understanding the nature of the population. For example, there are many social media and online communities. What kind of groups and links were posted? How was the study described and framed? Online groups devoted to sexual minority communities might disproportionately attract people who are in a stage of identity questioning, for example. Moreover, how were non-minority groups targeted (i.e., cis-hetero participants)?

More information should be provided about any missing data, whether any participants were omitted from the final sample, and any efforts taken to prevent or screen out duplicate cases or inattentive and careless respondents.

The “other” category for sexual identity was fairly large. What kinds of labels did people in this group use?

The tables report only the significant differences. Nevertheless, it is good reporting practice to report descriptive statistics for ALL the group comparisons. Given the high interest in these groups, I imagine that many researchers would be interested in many of the other variables. Moreover, null effects can be interesting in their own right. Expanding the current tables is worthwhile.

Finally, there is a minor mistake in the Results:

SCC score is a significant predictor (β = 0.066, p. < 0.05), which means that the higher the self-concept clarity, the higher the sense of coherence. (The wrong regression coefficient was reported; .653 was probably intended.)

Author Response

Thank you very much for your review and valuable remarks. The manuscript has been corrected as suggested.

At the beginning of the study the participants were given information about purpose, subject and course of the study and that participation in it was completely anonymous and voluntary. After getting acquainted with this information they agreed to participate in the study, which allowed access to further parts of the questionnaire. It would not be possible to complete other parts of the questionnaire without consent. All respondents agreed to participate in the study.

The study has been conducted using the Facebook platform. Link to the study has been shared on groups dedicated to topics such as psychology and sexology but also groups affiliated to the LGBTQ+ communities. Above mentioned groups are a space for their users to exchange views and thoughts. They also aim to support and integrate people from the LGBTQ+ community. In order to join such a group every member had to agree to the rules applicable in it. These groups are monitored by their administrators and moderators which have agreed to post the link  to the study on them. Only the willing people took part in the study. Each of the groups included cis-hetero individuals.

There was no need to delete any data. The design of the study made it impossible to submit the form without answering every question, as well as sending another answer from the same device, which made duplication unlikely to happen. The nature of the media used and the voluntary participation in the study, made it possible to avoid careless respondents.

Due to the large variety of answers and the need for simplification, the category ‘’Other’’ covers every person who, when specifying their sexual identity, went beyond the specified answer options (“Heterosexual”, “Homosexual”, “Bisexual”, “Asexual”, “I have not yet specifed my sexual orientation”) and used the space for the open answer. People who defined themselves as pansexual (24) and demisexual (8) created quite a large group. Some of the respondents answered the question about their sexual identity in a descriptive manner. In this group of respondents there were answers such as: ‘’I do not get attached to any psychosexual orientation, I do not want it to limit me.”, ‘’I have the impression that my orientation changes depending on my mood and I cannot explain it.’’, ‘’I am pretty sure that I am heterosexual, but I don’t exclude that I may be bisexual’’, ‘’At the moment I am close to asexuality but I am not 100% sure.’’. In addition, several people used 2 labels to define their sexual identity, for instance: ‘’bisexual and heteromantic‘’, ‘’pansexual and demisexual‘’, ‘’heterosexual and demisexual‘’, ‘’demisexual and sapiosexual”.  

Thank you also for your comments about the Results section. The results have been improved and the mistake has been amended. 

Reviewer 2 Report

An interesting read that explores the furthering insights with the 'self' and how the contemporary narrative is being challenged and critiqued. The complexity of such issues is all the more relevant for further research to be done. The findings have noted mixed results with some assumed correlations as well as some unexpected, showing the diverse understanding and perceptions that exist.         

Just a couple of minor suggestions:

- Maybe add in a copy of the Schwartz Circular Model for representation.

- Some of the phraseology within some paragraphs were convoluted, possibly a tweaking to become more accessible (less is more, more succinct comments etc.)  

Author Response

Thank you very much for your review and valuable comments.

We added the Schwartz Circular Model and the presentation of the result has been improved. Indicating which parts of the text regarding phraseology could be corrected would make it easier for us to make changes. The text has been checked by us in this regard.